# A mechanistic, stigmergy model of territory formation in solitary animals: Territorial behavior can dampen disease prevalence but increase persistence

**Lauren A. White** [1]*, **Sue VandeWoude** [2], **Meggan E. Craft** [3]

**1** National Socio-Environmental Synthesis Center, University of Maryland, Annapolis, Maryland, United States of America, **2** Department of Microbiology, Immunology & Pathology, Colorado State University, Fort Collins, Colorado, United States of America, **3** Department of Veterinary Population Medicine, University of Minnesota, St. Paul, Minnesota, United States of America

* lwhite@sesync.org

**Data Availability Statement:** Code is deposited on Zenodo (https://doi.org/10.5281/zenodo.3731357).

**Funding:** This work was supported by NSF DEB (1413925 and 1654609) and the National Socio

## Abstract

Although movement ecology has leveraged models of home range formation to explore the effects of spatial heterogeneity and social cues on movement behavior, disease ecology has yet to integrate these potential drivers and mechanisms of contact behavior into a generalizable disease modeling framework. Here we ask how dynamic territory formation and maintenance might contribute to disease dynamics in a territorial, solitary predator for an indirectly transmitted pathogen. We developed a mechanistic individual-based model where stigmergy—the deposition of signals into the environment (e.g., scent marking, scraping)—dictates local movement choices and long-term territory formation, but also the risk of pathogen transmission. Based on a variable importance analysis, the length of the infectious period was the single most important variable in predicting outbreak success, maximum prevalence, and outbreak duration. Host density and rate of pathogen decay were also key predictors. We found that territoriality best reduced maximum prevalence in conditions where we would otherwise expect outbreaks to be most successful: slower recovery rates (i.e., longer infectious periods) and higher conspecific densities. However, for slower pathogen decay rates, stigmergy-driven movement increased outbreak durations relative to random movement simulations. Our findings therefore support a limited version of the "territoriality benefits" hypothesis—where reduced home range overlap leads to reduced opportunities for pathogen transmission, but with the caveat that reduction in outbreak severity may increase the likelihood of pathogen persistence. For longer infectious periods and higher host densities, key trade-offs emerged between the strength of pathogen load, the strength of the stigmergy cue, and the rate at which those two quantities decayed; this finding raises interesting questions about the evolutionary nature of these competing processes and the role of possible feedbacks between parasitism and territoriality. This work also highlights the importance of considering social cues as part of the movement landscape in order to better understand the consequences of individual behaviors on population level outcomes.

Environmental Synthesis Center (SESYNC) under funding received from the NSF DBI 1639145. The funders had no role in study design, data collection and analysis, decision to publish, or preparation of the manuscript.

**Competing interests:** The authors have declared that no competing interests exist.

## Author summary

Making decisions about conservation and disease management relies on our understanding of what allows animal populations to be successful, which often depends on when and where animals encounter each other. However, disease ecology often focuses on the social behavior of animals without accounting for their individual movement patterns. We developed a simulation model that bridges the fields of disease and movement ecology by allowing hosts to inform their movement based on the past movements of other hosts. As hosts navigate their environment, they leave behind a scent trail while avoiding the scent trails of other individuals. We wanted to know if this means of territory formation could heighten or dampen disease spread when infectious hosts leave pathogens in their wake. We found that territoriality can inhibit disease spread under conditions that we would normally expect pathogens to be most successful: when there are many hosts on the landscape and hosts stay infectious for longer. This work points to how incorporating movement behavior into disease models can provide improved understanding of how diseases spread in wildlife populations; such understanding is particularly important in the face of combatting ongoing and emerging infectious diseases.

## Introduction

According to the general conceptual framework proposed by Nathan et al. [1], there are four motivating questions for movement ecology research: (1) why move?; (2) how to move?; (3) when and where to move?; and (4) what are the ecological and evolutionary consequences of moving? Recently, there has been a call for the discipline of movement ecology to better address the fourth component of this framework: the population-level consequences of moving [2]. In particular, researchers have argued for a greater synthesis of movement ecology with biodiversity [3] and disease ecology research [4,5]. One of the goals of incorporating such detail is to be able to observe the emergence of complex ecological and evolutionary processes that may depend upon individual traits like personality or behavioral phenotypes [6]. Pathogen transmission is one such process that is highly dependent on whether two conspecifics encounter each other within a certain window of time and space.

Mechanistic models of home range formation have their roots in a spatially-biased random walk process [7]. These models have evolved to incorporate underlying resource availability and selection, population dynamics, and territorial behaviors such as scent marking that lead to dynamic home range formation [8–13] resulting in individual interactions. Even so, disease ecology has yet to universally account for contact behavior that is driven explicitly by individual movement patterns [4,5,14]. Models in disease ecology are often specific to a given-host pathogen system or emphasize the risk of contact rather than ongoing transmission dynamics [5].

This disciplinary trajectory is problematic because wildlife vary in social organization on axes of gregariousness (group living vs. solitary) and territoriality (territorial vs. nonterritorial); each population structure has its own potential effects on pathogen transmission [15,16]. In an evolutionary context, parasites are a possible cost of group living, and host gregariousness is hypothesized to correlate with increased parasite prevalence, infection intensity, and parasite species richness [17,18]; however, this hypothesis lacks strong empirical support [17,18]. In particular, the relationship between group size and prevalence of parasitism may be confounded by host movement and territorial behavior [16]. A corollary to this idea is that

populations with smaller groups or spatially structured populations may be more protected from parasite transmission from external groups [19].

One possible mechanism for the maintenance of territories and spatial structure within populations is stigmergy. Stigmergy describes environmentally mediated feedback where the signals that one individual leaves in its path alter the behavior of its conspecifics, even after the individual has left that location [13]. In social insects, stigmergy helps to explain how individual pheromone trails can shape social organization of colonies [20]. In territorial animals, equivalent cues include marking through urine, scat, or community scrapes [21–24]. For example in puma (*Puma concolor*), males alter their visitation rates to community scrapes depending on the presence or absence of females or male competitors [25]. In a disease context, these non-contact territorial defense strategies (e.g., vocalization, scent marking, scraping) may have evolved to reduce transmission risk between individuals or groups [26,27]. Population thresholds are a key concept in epidemiology and disease ecology and lie at the root of disease control focused on reducing a susceptible population through culling or vaccination to reduce the likelihood of outbreaks [28]. Social and spatial structure, potentially mediated by such signaling, is one hypothesis for why population thresholds lack strong empirical support in wildlife populations [27–29].

Here we developed a generalizable mechanistic framework that examines the interplay between indirect pathogen transmission and dynamic territory formation motivated by deposition and response to signals left in the environment by hosts, i.e., stigmergy cues. We scale up the consequences of these individual decisions to simulate movements, interactions, and pathogen spread across a population. We then ask: (1) how do pathogens spread in populations responding to stigmergy stimuli (e.g., scent/territorial marking) compared to populations where individuals move randomly?; and (2) what are the consequences in trade-offs between strength and duration of scent mark vs. pathogen load and duration deposited in the environment? Here we explore the potential role of stigmergy not only in dynamic territory formation [12,30], but as a potential mitigator or facilitator of pathogen transmission in populations.

## Model

### Individual-based stigmergy movement model

We simulated stigmergy-driven and random movement for a closed population (no births, deaths, immigration, or emigration) [31] operating in discrete time and space. For both types of movement, individuals could move within a Moore neighborhood (eight neighboring cells) or remain within their current cell during each time step. For a landscape of $k = 1, \ldots, 9$ discrete grid cells, the probability of an individual moving from current location, $a$, to a new location, $b$, over a fixed temporal time step was:

$$P(a, b) = \frac{\phi(a, b)}{\sum_{k=1}^{9} \phi(a, c_k)}$$

Where $\phi(\cdot)$ is a 2D movement kernel and $c_k$ represents the center point of each grid cell. For the movement kernel, we assumed the simplest case of a uniform circular distribution:

$$\phi(r) = 1/(2\pi r^2)$$

where the movement kernel is inversely proportional to radial distance ($r$) from the center point of the current grid cell such that:

$$r = \sqrt{(x_a - x_c)^2 + (y_a - y_c)^2}$$

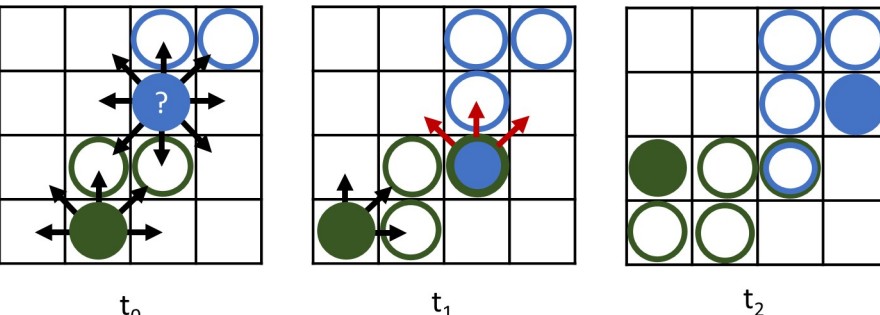

**Fig 1. Schematic for individual-based stigmergy model.** As hosts (solid circles) walk randomly through space they deposit scent marks (open circles); note the green vs. blue scent marks for each host. If infected, the hosts simultaneously leave pathogens in the environment. $t_0$: focal individual (blue) can move to one of eight neighboring cells or remain within its current cell. $t_1$: In the case of encountering a conspecific cue (open green circle), the host has some probability of constraining its next step to 45 degrees on either side of the prior direction of movement (indicated by the directions of the red arrows); this probability depends on the strength of the conspecific cue that the host encounters. $t_2$: avoiding conspecific stigmergy cues results in dynamic home range formation, but also potential pathogen exposure.

The equation gives an inverse distance weight (i.e., $1/r$) that is multiplied by the circumference at that distance to account for a uniform circular distribution (i.e., $1/(2\pi r)$)[32]. Note that the area under this kernel does not equal to one. By setting the minimum distance ($r$) for the cell of origin to 0.75, hosts were slightly more likely to remain in the current cell ($P = 0.23$) than move in one of the four cardinal directions ($P = 0.13$) or move in a diagonal direction ($P = 0.06$).

For stigmergy-driven movement, hosts navigated the landscape randomly based on this movement kernel, unless an individual encountered a scent marker from another individual during the previous time step (Fig 1, $t_0$). At each time step, every individual deposited a scent mark with initial intensity, $\eta_0$, at their current location (Fig 1). Scent mark strength in the environment decayed exponentially through time at rate $\delta$. Thus, the current scent mark strength at cell, $x$, and at time, $t$, was given by: $\eta(x, t) = \sum_{j=1}^{J} \eta_0 e^{-\delta(t-d_j)}$, where $d_j$ is the time of deposition by individual $j$ in a subset of the total population ($j = 1\ldots J$ individuals) that has visited cell $x$.

The hosts' movement responses to these stimuli depended on the strength of the scent load encountered. An individual's scent exposure was taken as: $\min(1, \eta(x,t))$, where $\eta(x,t)$ represents the sum of all active scent load deposited by all hosts in cell location $x$ on the landscape at time $t$. The subsequent direction of movement was determined by a Bernoulli trial: if $P < \min(1, \eta(x,t))$, the direction of movement was constrained to 45 degrees on either side of the direction of movement that brought that animal to the current cell (Fig 1, $t_1$). For example, a host encountering a foreign scent mark after moving to the bottom middle cell could move to the upper left, upper, or upper right cell from the current, scent marked cell (Fig 1, $t_1$-$t_2$). If $P > \min(1, \eta(x,t))$, the direction of movement was random, as described by the movement kernel above. This type of lattice model of territory formation results in dynamic territories that change through time (Fig 2) [12,13,33].

This framework is consistent with some previous simulation models, in that individuals move at random unless they encounter a foreign scent cue [12,13]. We have adapted these frameworks to evaluate impact of scent cues on pathogen transmission. Unlike prior models, movement could occur to diagonal cells, and response to scent cues was driven by the quantity of scent load which decays through time, rather than an explicit "active scent time" after which hosts no longer respond (per [12]). This mechanistic framework differs from prior models of

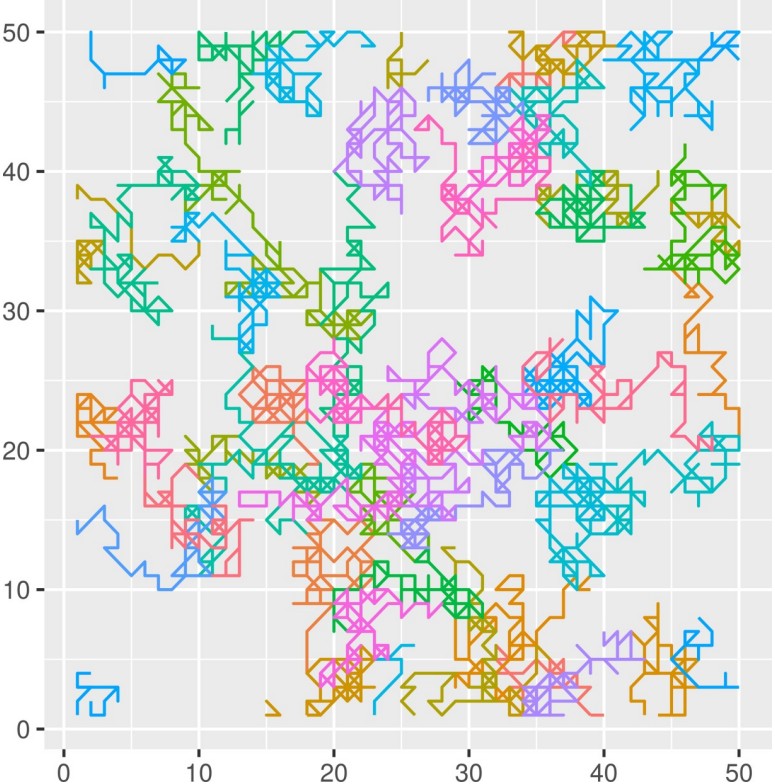

**Fig 2. Movement trajectories for simulated hosts.** Simulations occurred on a 50 x 50 landscape with wrapped edges (i.e., torus) to avoid boundary effects. Populations were simulated with 50, 100 or 150 individuals. Trajectories shown for a simulation with 50 hosts (a density of 0.02 hosts/unit$^2$), each represented by a different color, and a scent cue decay rate of 0.01 per time step for 100 time steps.

territory formation in that it assumes no directional bias, centralizing tendency, spatial auto-correlation, or increasing marking behavior in response to foreign scent cues [8,34,35]. We also did not consider responses to habitat, terrain, or resource availability [8,9,35]. Because we were investigating several dimensions of pathogen transmission, we simplified the formulation of the movement kernel to depend only on radial distance from the current location, unlike past models, which often assume distinct distributions of step length and directionality [35,36].

## Pathogen transmission process

We simulated the spread of an environmentally transmitted pathogen in a closed population using an SIR framework [31]. Infected individuals, in addition to leaving a scent mark, also deposited pathogens into the environment with intensity, $\kappa_0$. The pathogen load in the environment then decayed exponentially at a rate, $\alpha$. Pathogen load was cumulative—so if two infected individuals visited the same cell in sequence, the pathogen load in the environment reflected the sum of their two visits. Paralleling the scent mark decay process, the pathogen load intensity in a cell location, $x$, and time $t$ was given by: $\kappa(x, t) = \sum_{q=1}^{Q} \kappa_0 e^{-\alpha(t-d_q)}$, where the initial pathogen load, $\kappa_0$, decayed exponentially at rate $\alpha$ since the time of deposition, $d_q$, by each individual $q$ from a subset of the population ($q = 1\ldots Q$ infected individuals) that has visited cell $x$.

The probability of a susceptible individual becoming infected was governed by a Bernoulli trial where $\beta$ corresponds to the probability of a successful transmission event:

$$\beta = \min(1, \kappa(x, t))$$

where $\kappa(x,t)$, as defined above, is the sum of active pathogen load that has not yet decayed from all previous infected individuals visiting that cell. Thus, for a susceptible individual in a cell with environmental contamination, the transmission rate, $\beta$, is specific to the pathogen load, $\kappa$, remaining in the environment at time, $t$, in a particular cell, $x$. Infected hosts then have a probability, $\gamma$, of recovering per time step.

Like simulated movement, the disease transmission process occurred probabilistically, in discrete time, and on a spatially explicit landscape. However, a mean field approximation of the transmission process can be conceptualized as [37]:

$$\frac{dS}{dt} = -\beta(\kappa)S$$

$$\frac{dI}{dt} = \beta(\kappa)S - \gamma I$$

$$\frac{dR}{dt} = \gamma I$$

$$\frac{d\kappa}{dt} = \kappa_0 I - \alpha\kappa$$

where $S$ is the number of susceptible individuals, $I$ is the number of infected individuals, $R$ is the number of recovered individuals, and $\kappa$ is the total pathogen load in the environment. As outlined above, $\beta(\kappa)$ is a site-specific transmission probability dependent on the pathogen load at cell $x$ and at given time $t$, and infected individuals recover at a rate $\gamma$. Infectious individuals deposit $\kappa_0$ pathogen load into the environment which decays exponentially at rate, $\alpha$. The total population size ($N$) remains constant such that: $N = S+I+R$.

## Initial conditions, parameter space, and outcome metrics

We simulated a 50 x 50-cell landscape with wrapped edges (i.e., torus) to avoid boundary effects [38]. At the start of each simulation, individuals were randomly distributed across the theoretical landscape, and one individual was randomly selected to be infected, serving as the index case. We tested population sizes of 50, 100, and 150 individuals for respective host densities of 0.02, 0.04, and 0.06 hosts/unit area respectively. Since the transmission probability was controlled by the strength of pathogen load encountered in the environment rather than a fixed transmission probability, we explored the epidemiological parameter space by simulating low, medium, and high recovery rates (Table 1). We also explored the interplay of low, medium, and high deposition strengths for scent marking and pathogen shedding, as well as low, medium, and high rates of decay for pathogen infectiousness and scent mark strength (Table 1). Finally, we compared stigmergy-driven, territorial simulations with their random movement counterparts ($m$). In total, we tested 1,458 parameter sets with 100 simulations per parameter set (Table 1). For each parameter set, we recorded mean outbreak success (did the disease spread beyond the initial index case?), mean maximum prevalence, and mean outbreak duration (the number of time steps until there were no remaining infectious individuals on the landscape). We also used a random forest variable importance analysis to assess the relative importance of each parameter on these three outcomes.

**Table 1. Factorial design of 1,458 parameter combinations encompassing host density, recovery rate, pathogen load and decay rate, and scent load and decay rate.**

| Parameter | Values tested | Description |
|---|---|---|
| $N$ | 50, 100, 150 hosts/50 units$^2$ = 0.02, 0.04, 0.06 hosts/unit$^2$ | Host density |
| $\gamma$ | 0.01, 0.05, 0.10 time$^{-1}$ | Recovery rate |
| $m$ | TRUE, FALSE | Stigmergy driven (T) or random movement (F) |
| $\kappa_0$ | 0.1, 1, 10 | Pathogen load: initial strength of pathogen load deposited into the environment |
| $\eta_0$ | 0.1, 1, 10 | Scent load: Initial strength of scent mark deposited into environment |
| $\alpha$ | 0.01, 0.1, 1 | Pathogen decay rate: exponential decay rate of pathogen infectiousness in environment |
| $\delta$ | 0.01, 0.1, 1 | Scent decay rate: exponential decay rate of scent mark strength |

## Variable importance analysis

We explored model sensitivity to parameter values by conducting a random forest variable importance analysis. Random forest analysis is an approach that accounts for non-linear and collinear relationships between variables, allows for different variable types (e.g., numerical vs. categorical), and avoids the concerns of using frequency-based statistical p-values to assign significance in a simulation context [39,40]. Random forest analysis generates an ensemble of classification or regression trees for a given data set and then combines predictions from the individual trees [39]. Variable importance results are reported as mean decrease in accuracy scores, which describes the loss in accuracy to the predicted outcome when the given variable is permutated randomly [39]. We used the *randomforest* function to generate of 1,000 trees for the metrics of outbreak success (did the pathogen spread beyond the initially infected individual?), maximum prevalence given outbreak success, and outbreak duration given outbreak success. With 1,000 trees, the order of variable importance did not switch with different random seeds and the error rate or mean squared error of the random forest stabilized. We further evaluated our model performance using separate training (80% of data) and test (20% of data) data sets. Error rates on training data sets were less than 30%, and model accuracy on test data sets exceeded 70% across the three outcomes (S1 Table). Finally, we also verified our approach with the *party* package, which has been shown to be particularly robust to bias relative to the traditional *randomforest* package [41–43]. Overall, order of variable importance order was robust to using *randomforest* vs. *cforest* approaches. All simulations and analyses were conducted in R (version 3.5.3). Code is deposited at Zenodo (doi.org/10.5281/zenodo. 3731357).

## Results

### Recovery rate critical in spread of indirectly transmitted pathogens

The random forest variable importance analysis indicated that recovery rate ($\gamma$) was the single most important variable in predicting the probability of a successful outbreak, maximum prevalence, and outbreak duration (Fig 3). Host density ($N$) and decay rate of pathogen infectiousness ($\alpha$) followed as the next most important variables for predicting all three outbreak metrics (Fig 3). However, for maximum prevalence specifically, pathogen decay rate ($\alpha$) slightly exceeded host density ($N$) in variable importance (Fig 3B). Whether or not an outbreak had stigmergy-driven vs. random movement ($m$) had little impact on whether or not an outbreak

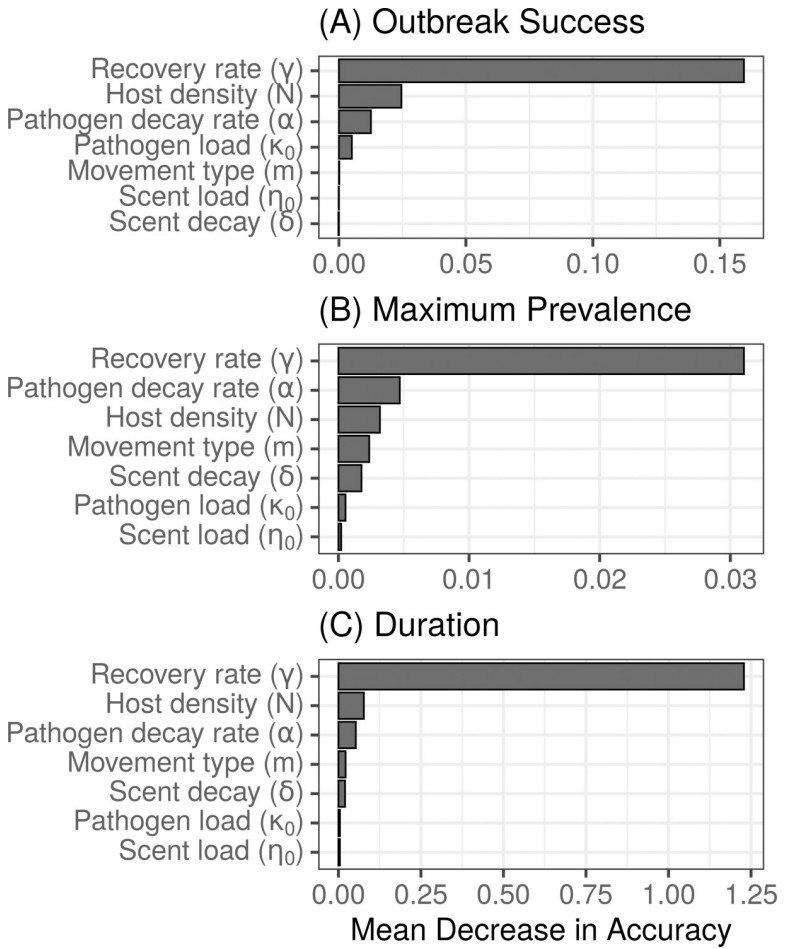

**Fig 3.** Random forest analysis for: (A) outbreak success (did the pathogen spread beyond the initially infected individual?), (B) maximum prevalence given outbreak success, and (C) outbreak duration given outbreak success.

took place (Fig 3A), but did contribute to determining the maximum prevalence and duration of successful outbreaks (Fig 3B and 3C). Outbreaks with faster recovery rates (i.e., 0.1 and 0.05 per time step) had a lower maximum prevalence and shorter outbreak durations regardless of whether movement was random or driven by stigmergy cues.

## Territoriality can reduce outbreak severity, but increase disease persistence

Territorial movement yielded a lower maximum prevalence in scenarios that were already conducive to outbreaks: a higher host density and slower recovery rates (i.e., longer infectious periods). These mitigating effects were strongest for simulations with higher host densities, yielding a median reduction in maximum prevalence of 0.05–0.10 relative to random simulations with equivalent epidemiological parameter sets (Fig 4A). These reductions in maximum prevalence decreased to ~0.05 and ~0.025 for lower host densities (S1 and S2 Figs). In contrast, for the highest host density, stigmergy increased outbreak duration relative to random movement, most notably for simulations with slower pathogen decay rates (Fig 4B). For lower host densities and slower recovery rates, an interaction emerged between pathogen decay rate and movement type (S1 and S2 Figs). For the lowest host density and slower recovery rates, stigmergy driven movement increased outbreak duration when pathogen decay rates were slower, but decreased outbreak duration for faster pathogen decay rates (S2 Fig).

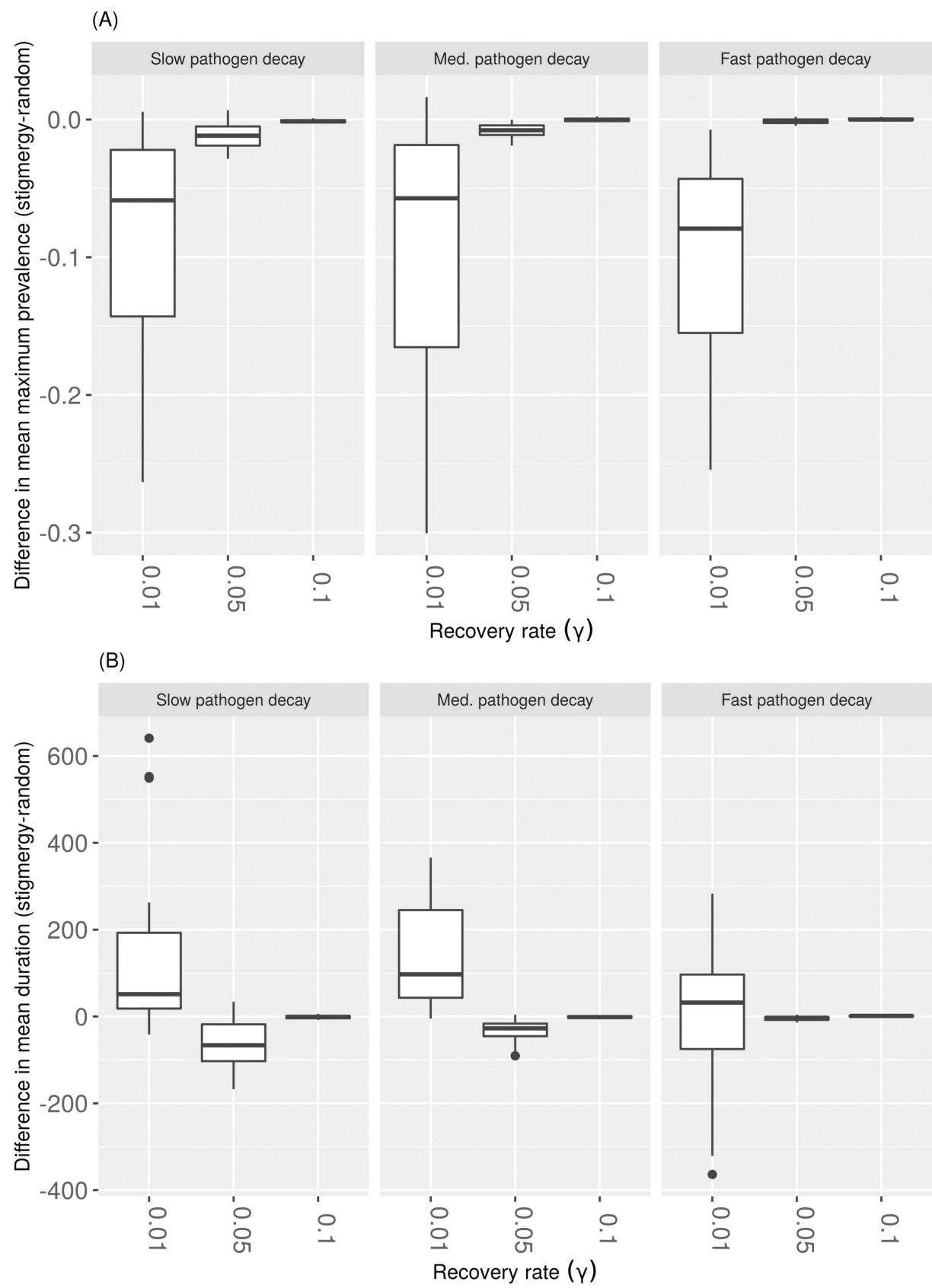

**Fig 4. The absolute difference between stigmergy vs. random movement simulations for (A) mean maximum prevalence and (B) mean outbreak duration as a function of recovery rate and environmental decay rate of pathogen ($\alpha$, columns).** Each point in the box plot distribution represents a paired difference between the mean outcomes for stigmergy vs. random simulations for a given parameter set. Shown for the highest host density of 0.06 hosts/unit$^2$.

## Non-linear interactions between pathogen load, pathogen decay, scent load, and scent decay

In the parameter space where outbreaks were most successful (e.g., slower recovery rates and higher host densities), non-linear patterns emerged from interactions between decay rate of pathogen infectiousness, decay rate of scent cue, initial pathogen load, and initial strength of scent cue. With both the highest host density and the slowest recovery rate ($\gamma = 0.01$), outbreaks reached a higher maximum prevalence for simulations with higher initial pathogen loads, slower pathogen decay rates, lower initial scent loads, and faster scent decay rates (Fig 5A and S3 Fig, lower left quadrant). In contrast, outbreaks lasted longer on average for simulations with higher initial pathogen loads, slower pathogen decay rates, but higher initial scent loads, and slower scent decay rates (Fig 5B and S3 Fig, upper left quadrant). These trends weakened with lower host densities (S4 Fig). However, at higher host densities with intermediate recovery rates ($\gamma = 0.05$), slow pathogen decay, fast scent decay, and high initial pathogen and scent loads favored longer outbreaks (S5 and S6 Figs). These patterns dissolved for faster recovery rates and lower host densities where outbreaks were less successful (S7–S10 Figs).

For simulations with slower recovery rate and higher host density, response to initial scent load was variable: high initial scent load promoted outbreaks under slow pathogen decay, but inhibited outbreaks under conditions of faster pathogen decay (Fig 6 and S11 Fig). Together, fast pathogen decay and fast scent load decay rates were not conducive to outbreaks regardless of initial pathogen load or scent load (Fig 6 and S11 Fig). Fast scent decay and slow pathogen decay also minimized the effect of different initial pathogen and scent loads (Fig 6 and S11 Fig). Lower host density treatments increased variability across outcomes and minimized the differences across scent decay rate, initial pathogen load, and initial scent load for a given pathogen decay rate (S12 Fig).

## Discussion

Adaptive, dynamic, or territorial space use is one possible explanation for the lack of empirically observed density thresholds in wildlife [27,28]. This "territoriality benefits" hypothesis suggests that reduced home range overlap could lead to reduced opportunities for pathogen transmission [44]. Our findings support this hypothesis with the caveat that a reduction in outbreak severity may come at the cost of increased likelihood of persistence for indirectly transmitted pathogens. We found that territoriality did indeed reduce maximum prevalence of disease in conditions where we would otherwise expect outbreaks to be most successful: slower recovery rates (i.e., longer infectious periods) and higher conspecific densities (Fig 4 and S1 Fig). However, for higher host densities, outbreak duration decreased for populations with stigmergy-driven movement compared to their randomly moving counterparts (Fig 4). Interestingly, at lower host densities, an interaction emerged with pathogen decay rate; stigmergy-driven movement could increase outbreak duration times when pathogen decay rates were faster (S2 Fig).

For longer infectious periods and higher host densities, key trade-offs emerged between the strength of pathogen load, strength of the stigmergy cue, and the rate at which those two quantities decayed. Intuitively, high initial pathogen load and a slower pathogen decay rate universally promoted higher maximum prevalence and longer lasting outbreaks (Figs 5 and 6). In

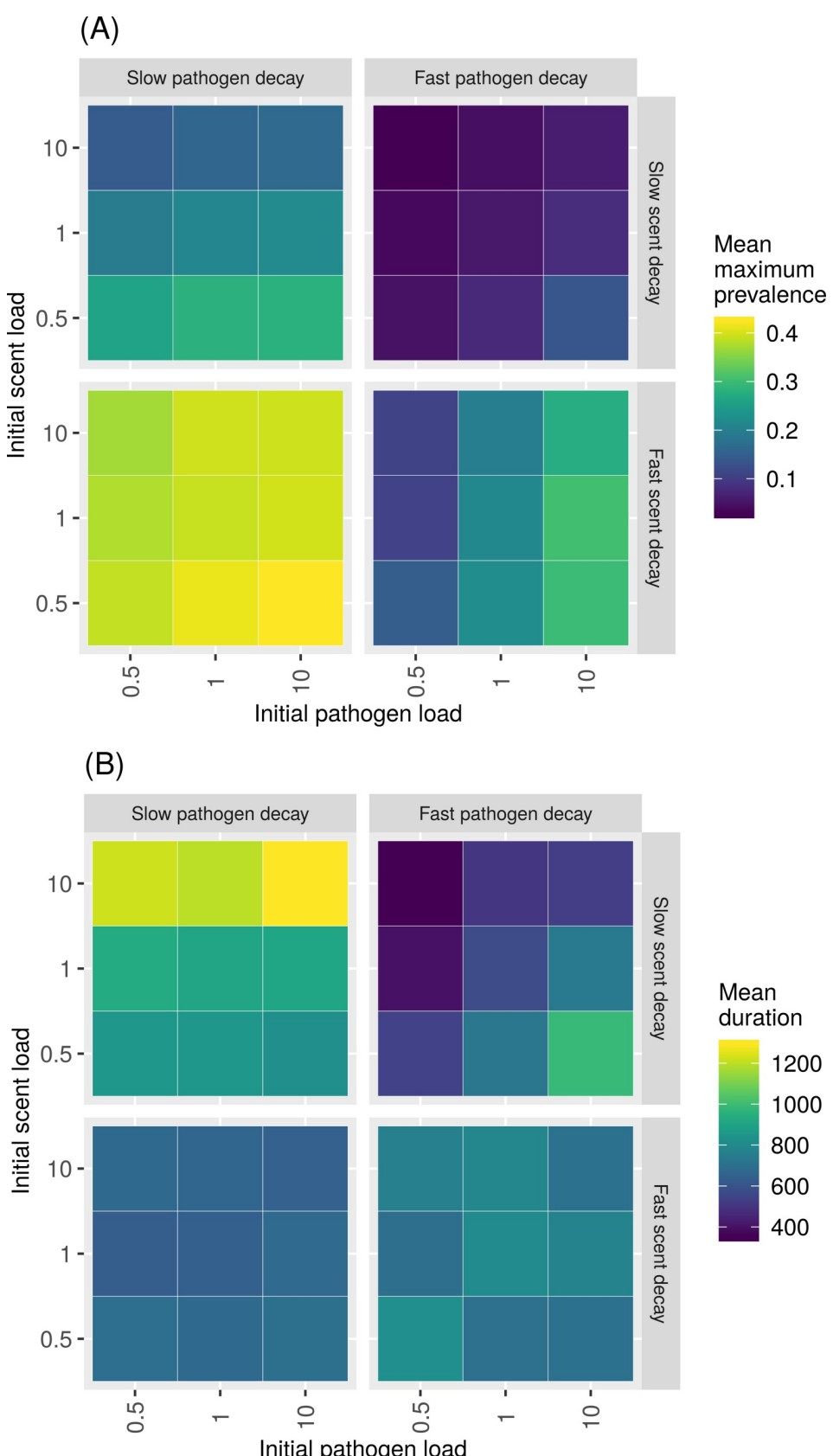

**Fig 5.  Mean maximum prevalence (A) and mean duration (B) of simulated outbreaks for simulations with a high host density (0.06 hosts/unit$^2$) responding to stigmergy cues with a recovery rate of 0.01/unit time.** Columns correspond to pathogen decay rates ($\alpha$) while rows correspond to scent decay rates ($\delta$).

contrast, lower initial scent loads paired with faster scent decay promoted higher maximum prevalences (Fig 5A), whereas lower initial scent loads and slower scent decay rates promoted longer lasting outbreaks (Fig 5B). These findings raise interesting questions about the evolutionary nature of the competing strength of pathogen and scent marking signals in the environment. For indirectly transmitted pathogens, pathogens should coevolve for longer persistence and higher virulence because individual host mortality is less important to a pathogen's overall fitness [45]. This is in opposition to the prediction that populations with spatially restricted movement will contribute to the evolution of less virulent pathogens [45]. Our results support the idea that pathogens co-opting their hosts' social communication system could help to overcome territorial barriers (e.g., [46]) and that territorial behavior could offer benefits for stochastic persistence from a pathogen's evolutionary perspective (Fig 4). In particular, a host's tendency to deposit less strong, but more slowly decaying scent mark could help maintain pathogen persistence (Fig 5B).

The relationship between population thresholds and indirectly transmitted pathogens remains an open question [28]. Feedbacks between host behavior and parasitism are likely to complicate this relationship further. Hosts have evolved defenses and avoidance behaviors in response to high parasitism risk (e.g., altering of ranging patterns in primates or selective foraging with behavioral avoidance of fecal-contaminated areas in ungulates) [16]. However, pathogens may have co-evolved to counteract territorial barriers and exploit social signaling behaviors. Some preliminary evidence suggests that this may occur. For example, wild banded mongooses (*Mungos mungo*) transmit the mycobacterium, *M. mungo*, almost solely through anal and urine secretions, which are key currencies in their social communication system [46]. Similarly, higher rates of raccoon roundworm (*Baylisascaris procyonis*) infection occur at latrine sites compared to individual raccoon sites; this could lead to higher infection rates for susceptible raccoons and intermediate hosts attracted to undigested seeds [47]. While there is preliminary empirical evidence about the potential role of social signaling behaviors in indirect pathogen transmission, we lack a clear understanding of whether stigmergy is a potential mitigator or facilitator of pathogen transmission at a population level.

In our model, indirectly transmitted pathogen dynamics could be altered by territorial cues if hosts existed at high enough densities and shed pathogens for long enough across the landscape. This work, therefore, highlights the importance of exploring feedbacks between territoriality and parasitism. In empirical systems, hosts with lower levels of parasitism may be better able to form and maintain territories. For example, pheasants are a competent host for Lyme disease and are commonly parasitized by *Ixodes ricinus* ticks. Male pheasants with experimentally reduced tick loads were more likely to gain harems and have smaller territories. In contrast, males with higher tick loads ranged more broadly in peripheral woods and fields leading to a positive feedback loop of higher likelihood of tick exposure [48]. Examples of negative feedback between parasitism and territoriality also exist. In male Grant's gazelle (*Nanger granti*), territorial behavior drives higher parasite loads, but higher parasite loads suppress behaviors associated with territoriality [49]. Future model development might consider incorporating such feedback mechanisms [50], e.g., differences in movement behavior between symptomatic and healthy individuals.

To highlight the competing axes of stigmergy cue strength and duration vs. pathogen load strength and duration, we simulated movement using a random walk rather than

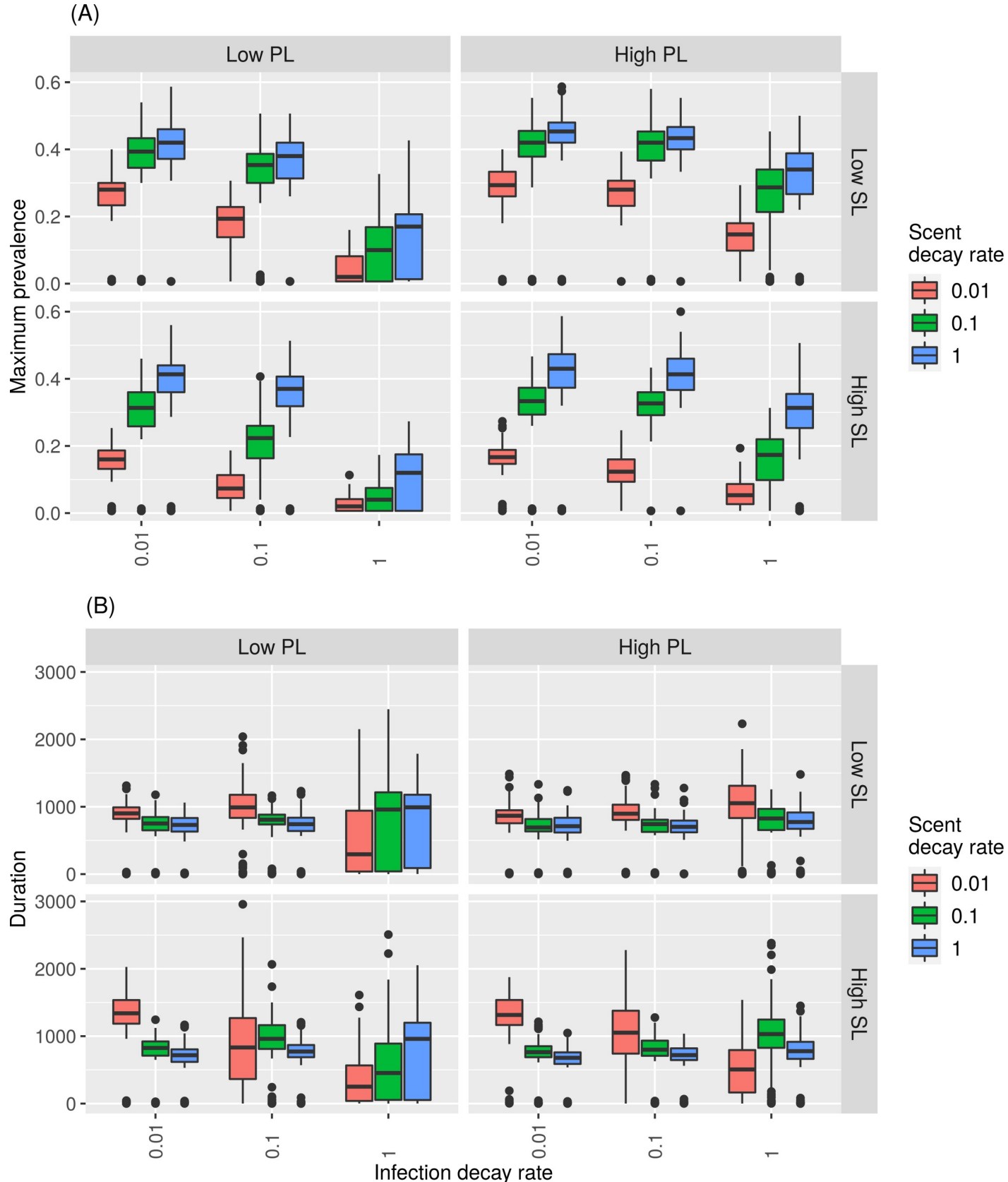

**Fig 6. Boxplots of (A) maximum prevalence and (B) outbreak duration for high host density (0.06 hosts/unit$^2$) responding to stigmergy cues and a recovery rate of 0.01/time step.** Rows correspond to low and high scent loads (SL). Columns correspond to low and high pathogen loads (PL).

incorporating additional potential complexities of movement behavior; this necessarily means that simulated individuals did not respond to the real-time presence or absence of conspecifics in neighboring cells. Future modelling studies could explore the sensitivity of results to differences in perceptual range (i.e., extending beyond a Moore neighborhood) and memory of past movements or past stigmergy cue encounters. Other extensions might include accounting for dispersal behavior or inter-individual differences in home range size. Ultimately, stigmergy is just one possible mechanism for informing territorial-like movement behavior. It is likely that many species respond to cues in real time (e.g., visual cues, vocalization) in addition to transient environmental cues (e.g., [11]). Another important question is understanding how temporal switches in the valence of the stigmergy cues might affect pathogen transmission. For example, during mating seasons scent cues could become attractive rather than aversive [21]. Individuals are also likely to display heterogeneous responses to different members of the population (e.g., male vs. female) and their environmental cues [51].

The model presented here best describes indirect or environmental transmission of a single infectious agent within a solitary, territorial host species. However, this model could also describe the behavior of social, territorial carnivores (e.g., gray wolves, African lions), where the movement of a single individual is usually representative of the entire group [52]. This model framework may also be relevant for pathogens with other dominant transmission modes that persist in the environment for extended periods. For example, canine parvovirus, which can persist up to one year outside of a host [53], is of conservation concern for wild carnivores [54]. Similarly, leptospirosis, a bacterial infection of wildlife (and humans), can persist for months in aqueous environments [55]. Small mammals, including peri-domestic species like raccoons, secrete bacteria through urine [56,57], which can serve as a scent marking communication tool [58]. Likewise, feline calicivirus remains infectious for up to 20 days at ambient temperatures [59] and is of epidemic concern for African lions [19]. Some domestic cats remain persistently infected (shedding virus for more than 30 days) and may shed higher levels of virus [17], which corresponds to the slower recovery rate condition in our model.

In an applied context, scent marking behavior can serve as a way to assess animal populations through time and document responses to human disturbance [46,60]. Our results support the idea that decision makers should evaluate possible changes in scent marking behavior and its potential effects on disease control when considering culling or altering population size in territorial species [61,62]. For example, prior attempts to control bovine tuberculosis (*Mycobacterium bovis*) in badgers through culling caused changes in scent marking behavior. At lower densities, badgers were more likely to have dispersed patterns of fecal and urine scent marking with higher concomitant risks of pathogen transmission [63]. Bovine tuberculosis transmission is thought to occur primarily through direct contact, but its ability to persist in the environment has raised questions about the role of indirect transmission routes [64,65]. Likewise, wildlife scent marking behavior at the human-livestock interface is of concern since some wildlife species (e.g., foxes, badgers) preferentially use farm food storage buildings for foraging and scent-marking which heightens pathogen transmission risk [66,67]. Scent marking may also influence the success of species reintroductions and population management: introducing translocated animals into an established territorial population may increase transmission risk because of increased overlap in home ranges or direct contacts [68]. Similarly, anthropogenic resource supplementation increases risk of indirect/fomite transmission (e.g., bTB in deer, brucellosis in elk, chronic wasting disease in deer and elk) [69]. An interesting

question moving forward would be to investigate the competing roles of habitat quality and territoriality on disease dynamics for pathogens with environmental persistence [32].

Existing movement ecology studies have so far focused on how to model territorial behavior and not the consequences of dynamic territories on population-level outcomes like disease [2]. This work provides a key interface between the disciplines of movement and disease ecology [4,5,14] by exploring how mechanistic movement driven by an individual's social landscape affects disease dynamics. These results indicate an interesting threshold at higher host densities where stigmergy-driven movement behavior can still support pathogen persistence. This framework can be adapted to specific host–pathogen systems to generate hypotheses about the competing roles of transient social cues and indirect pathogen transmission. We hope that this model inspires additional research surrounding the role of socially driven movement behavior and its concomitant implications for pathogen transmission.

## Supporting information

**S1 Table. Error rate and model accuracy from random forest models for three measured outcomes of disease dynamics.**
(DOCX)

**S1 Fig. The absolute difference between stigmergy vs. random movement simulations for (A) mean maximum prevalence and (B) mean outbreak duration as a function of recovery rate and environmental decay rate of pathogen ($\alpha$, columns).** Each point in the box plot distribution represents a paired difference between the mean outcomes for stigmergy vs. random simulations for a given parameter set. Shown for a medium host density of 0.04 hosts/unit$^2$.
(TIF)

**S2 Fig. The absolute difference between stigmergy vs. random movement simulations for (A) mean maximum prevalence and (B) mean outbreak duration as a function of recovery rate and environmental decay rate of pathogen ($\alpha$, columns).** Each point in the box plot distribution represents a paired difference between the mean outcomes for stigmergy vs. random simulations for a given parameter set. Shown for a low host density of 0.02 hosts/unit$^2$.
(TIF)

**S3 Fig. Mean maximum prevalence (A) and mean duration (B) of simulated outbreaks for simulations with a medium host density (0.04 hosts/unit$^2$) responding to stigmergy cues with a recovery rate of 0.01/unit time.**
(TIF)

**S4 Fig. Mean maximum prevalence (A) and mean duration (B) of simulated outbreaks for simulations with a low host density (0.02 hosts/unit$^2$) responding to stigmergy cues with a recovery rate of 0.01/unit time.**
(TIF)

**S5 Fig. Mean maximum prevalence (A) and mean duration (B) of simulated outbreaks for simulations with a high host density (0.06 hosts/unit$^2$) responding to stigmergy cues with a recovery rate of 0.05/unit time.**
(TIF)

**S6 Fig. Mean maximum prevalence (A) and mean duration (B) of simulated outbreaks for simulations with a medium host density (0.04 hosts/unit$^2$) responding to stigmergy cues with a recovery rate of 0.05/unit time.**
(TIF)

**S7 Fig. Mean maximum prevalence (A) and mean duration (B) of simulated outbreaks for simulations with a high host density (0.06 hosts/unit$^2$) responding to stigmergy cues with a recovery rate of 0.10/unit time.**
(TIF)

**S8 Fig. Mean maximum prevalence (A) and mean duration (B) of simulated outbreaks for simulations with a medium host density (0.04 hosts/unit$^2$) responding to stigmergy cues with a recovery rate of 0.10/unit time.**
(TIF)

**S9 Fig. Mean maximum prevalence (A) and mean duration (B) of simulated outbreaks for simulations with a low host density (0.02 hosts/unit$^2$) responding to stigmergy cues with a recovery rate of 0.05/unit time.**
(TIF)

**S10 Fig. Mean maximum prevalence (A) and mean duration (B) of simulated outbreaks for simulations with a low host density (0.02 hosts/unit$^2$) responding to stigmergy cues with a recovery rate of 0.10/unit time.**
(TIF)

**S11 Fig. Boxplots of (A) maximum prevalence and (B) outbreak duration with a medium host density (0.04 hosts/unit$^2$) responding to stigmergy cues and a recovery rate of 0.01/time step.** Rows correspond to low, medium, and fast scent loads (SL). Columns correspond to low, medium, and fast pathogen loads (PL).
(TIF)

**S12 Fig. Boxplots of (A) maximum prevalence and (B) outbreak duration with a low host density (0.02 hosts/unit$^2$) responding to stigmergy cues and a recovery rate of 0.01/time step.** Rows correspond to low, medium, and fast scent loads (SL). Columns correspond to low, medium, and fast pathogen loads (PL).
(TIF)

## Acknowledgments

The authors would like to thank Matthew Michalska-Smith for his review of the mathematical notation of this model.

## Author Contributions

**Conceptualization:** Lauren A. White, Sue VandeWoude, Meggan E. Craft.

**Data curation:** Lauren A. White.

**Formal analysis:** Lauren A. White.

**Funding acquisition:** Lauren A. White, Sue VandeWoude, Meggan E. Craft.

**Methodology:** Lauren A. White.

**Visualization:** Lauren A. White.

**Writing – original draft:** Lauren A. White.

**Writing – review & editing:** Lauren A. White, Sue VandeWoude, Meggan E. Craft.

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
