## [Decision Letter · Decision Letter 0]

10 Dec 2019

Dear Dr White,

Thank you very much for submitting your manuscript, 'A mechanistic, stigmergy model of territory formation in asocial animals: Territorial behavior can dampen disease prevalence but increase persistence', to PLOS Computational Biology. As with all papers submitted to the journal, yours was fully evaluated by the PLOS Computational Biology editorial team, and in this case, by independent peer reviewers. The reviewers appreciated the attention to an important topic but identified some aspects of the manuscript that should be improved.

We would therefore like to ask you to modify the manuscript according to the review recommendations before we can consider your manuscript for acceptance. Your revisions should address the specific points made by each reviewer and we encourage you to respond to particular issues Please note while forming your response, if your article is accepted, you may have the opportunity to make the peer review history publicly available. The record will include editor decision letters (with reviews) and your responses to reviewer comments. If eligible, we will contact you to opt in or out.raised.

- Supporting Information uploaded as separate files, titled 'Dataset', 'Figure', 'Table', 'Text', 'Protocol', 'Audio', or 'Video'.

We hope to receive your revised manuscript within the next 30 days. If you anticipate any delay in its return, we ask that you let us know the expected resubmission date by email at ploscompbiol@plos.org.

Sincerely,

Matthew (Matt) Ferrari

Associate Editor

PLOS Computational Biology

Virginia Pitzer

Deputy Editor

PLOS Computational Biology

[LINK]

I thank the authors for their manuscript. Both reviewers have made helpful suggestions for changes that I believe will improve the clarity of the presentation. In particular:

R1 has asked for some clarification in the presentation of absolute differences in model outputs as a function of random vs. stigmergy movement. Further, R1 has asked for more evidence in support of the argument that "territoriality “may increase the likelihood of pathogen persistence” ". As I agree with the reviewer that this is an interesting conjecture, I would encourage the authors to make every attempt to strengthen the support for this claim using the evidence presented.

R2 has offered a number of suggestions for links to the existing literature that the authors should carefully consider. Further, I agree with R2 that the inclusion of the material in L141-204 in "results" is awkward. While we're at the mercy of the PLoS template, I have handled many papers that include a model description section between the introduction and results; I would recommend such a reorganization here. Further, I encourage the authors to clarify the transmission model per the request of R2.

Reviewer's Responses to Questions

**Comments to the Authors:**

Reviewer #1: In this manuscript, White et al. develop an individual based model that describes the dynamics of an environmentally transmitting pathogen through a population of asocial species where individuals display territorial movement. The main conclusion of the manuscript is that the duration of infectious period, population density and pathogen decay rate are the most important factors that influence outbreak severity, which is to be expected. Territoriality only influences outbreak severity when the conditions are conducive to outbreaks. For these scenarios, the authors report that territoriality reduces peak prevalence and may “increase the likelihood of pathogen persistence”. The model developed in this paper is interesting and important. It can be possibly be expanded in future to explore the impact of social behavior by modeling movement that is in between random and asocial territorial (but, ofcourse, this is beyond the scope of the current manuscript). I have a few concerns that I would like to see better caveated in the manuscript or addressed with changes to the analysis if feasible.

1) Analysis of the impact of territoriality on outbreak severity: Much of the results about the effects of territoriality relies on qualitative difference between pooled results for random and stigmergy movement (for example difference between top and bottom panel of Figure 4). I wonder if these results can be strengthened by instead reporting delta maximum prevalence and delta outbreak duration (where delta maximum prevalence = maximum prevalence (stigmergy movement) - maximum prevalence (random movement) for the same parameter combination). This approach would allow direct one-to-one comparison of how territorial movement would influence outbreak dynamics under similar parameter combination.

2) I also worry about the statement (in abstract and rest of the manuscript) that territoriality “may increase the likelihood of pathogen persistence” follows the finding that territoriality, under certain conditions makes the “outbreak duration considerably more variable”. Increase in variation around the mean does not imply that likelihood of pathogen persistence is higher. In fact, increase in variation implies that it is equally likely that the likelihood of pathogen persistence may be reduced. I believe that my previous suggestion of reporting values for delta outbreak duration will help to evaluate whether territoriality tends to increase or decrease outbreak duration.

3) Variable importance analysis: was the model performance evaluated using test data set? What was the prediction error of the model? Please show these results.

4) Line 229 – 236: Effect of stigmergy on outbreak duration: The authors report that territorial movement increases disease persistence in certain cases. However, looking at Figure 4b, it seems that there are only a handful of outlier cases were outbreak duration due to stigmergy is higher compared to random movement (please also see my comment on this issue above).

Minor comments:

This is a minor point, but using “population size” and “population density” for the same parameter is confusing. I suggest that the authors stick to using one terminology throughout the manuscript.

Discussion, line 289: I believe it should be “lower initial scent load … promoted higher maximum prevalences” instead of “higher initial scent load”

Line 212: (Fig 3B&C) instead of (Fig 3A&B).

Reviewer #2: This paper addresses the important topic of movement-driven disease transmission in a population of territorial animals. It combines a general individual-based model of conspecific-avoidant territorial movement with an SIR model. While its findings were not surprising, the model is valuable for its explicit integration of disease and behavioral dynamics. However, a lot of mechanistic details are left unexplained in the manuscript, particularly the assumptions made about the transmission process. In addition, there is a notable gap in the citations of relevant literature; critically, it overlooks several recent key theoretical works that are integral to the scope of the present study. Lastly, the manuscript needs to be substantially reorganized – currently, majority of the method description, which is also oddly attached at the end, is relegated to Results. All these problems could be addressed in the revision.

Major problems:

1. The model seems to be heavily influenced by recent mechanistic territory models by Potts, Giuggioli et al. However, it has significantly simplified their descriptions of the movement process, e.g. assuming no directional bias, centralizing tendency, or spatial autocorrelation. The formulation of the movement kernel also deviates from many past models, which often assume distinct distributions of step length and directionality (see Moorcroft and Lewis 2006). Clarifying these differences in the method section would to help put this work in context. More importantly, the entire mathematical description of the SIR model is missing from the main text. It includes essential information that would drastically alter the predictions, e.g. how density-dependent transmission is modeled. In other words, more equations need to be shown.

2. The paper's organization is extremely confusing. In the Results section, everything from lines 141-204 belongs in Methods, which should also include lines 372-406. Each major figure (Figs 4-6) has 12-18(!) panels, which is way too complex. I strongly recommend developing alternative visualization that compresses the most essential information down to at most 4 panels per figure. Moreover, the effects of population density on movement-dependent disease spread is an especially interesting problem that could be more extensively explored here.

Minor problems:

1. L38 – mechanistic models of animal territoriality, or more generally, home range formation, have been developed since the 80s (see Okubo 1980). See below for some recent works relevant to this manuscript.

2. L39 – collective movement is more commonly considered as a subset of movement ecology.

3. L41 – even exclusive territoriality does not equate to asociality; territorial maintaince typically requires direct or indirect conspecific interactions, which all involve social responses.

4. L63-64 – this is not accurate; plenty of empirical and theoretical works suggest otherwise

5. L77 – edit: “According to the general conceptual framework of movement ecology proposed by Nathan et al”

6. L88 – Not accurate; notably, see Moorcroft et al. (Proc B 2006), Wang and Grimm (Ecol Modelling 2007), Bateman et al. (J Animal Ecology 2015), Tao et al. (American Naturalist, 2016), and Potts et al. (Theoretical Ecology, 2019).

7. L99 – edit: “lacks strong empirical support”

8. L112-115 – I suggest reordering the sentences

9. L116-129 – population thresholds and host-pathogen coevolution are very much outside the scope of the present study; this material should be moved to the Discussion.

10. L163 – this constraint only occurs probabilistically according to L172

11. L169 – don’t italicize min

12. L172 – italicize p and all mathematical variables

13. L193 – this part of the model wasn’t discussed earlier

14. L195 – so an outbreak that only affects two individuals before burning out would count as a success?

15. L197 – why only model max prevalence, not mean? Justification needed.

16. Fig 3 – subtitles needed (i.e. outbreak likelihood, maximum prevalence, outbreak duration); spell out the variable names on the y axis and keep the same order for all three panels for visual clarity; remove %IncMSE from x label.

17. L225-226 – confusing wording

18. L385 – edit: “The probability of an individual relocating from x’ to x is determined by a movement kernel …”; also, note that the area under this kernel does not equal to one. Use conventional mathematical grammar in this paragraph (use e.g. Moorcroft and Lewis 2006, Bateman et al. 2015, as templates)

19. L389 – why a uniform distribution instead of a Gaussian or the conventionally used Von Mises distribution?

20. L403 – what are “trees” in this context?

**Have all data underlying the figures and results presented in the manuscript been provided?**

Reviewer #1: Yes

Reviewer #2: None

PLOS authors have the option to publish the peer review history of their article (what does this mean?). If published, this will include your full peer review and any attached files.

Reviewer #1: No

Reviewer #2: No

---

## [Decision Letter · Decision Letter 1]

1 Mar 2020

Dear Dr White,

Thank you very much for submitting your manuscript "A mechanistic, stigmergy model of territory formation in solitary animals: Territorial behavior can dampen disease prevalence but increase persistence" for consideration at PLOS Computational Biology. As with all papers reviewed by the journal, your manuscript was reviewed by members of the editorial board and by several independent reviewers. The reviewers appreciated the attention to an important topic. Based on the reviews, we are likely to accept this manuscript for publication, providing that you modify the manuscript according to the review recommendations.

I thank the authors for their careful attention to the reviewer comments. Based on the comments of R2 I would suggest a few minor changes (detailed below) before recommending this manuscript for publication. With these changes made, I see no reason that this will need to be sent back out to review and I will be happy to direct the manuscript for publication. Specifically:

R2 raises a question about the notation on L216. I assume, from the text, that the sum described in that line is over individuals. However, R2 found that the description left some doubt. Could you please clarify this notation. Perhaps (a suggestion, not a requirement) adding an index j in 1...J to k_x,y to indicate the you are summing the decaying pathogen loads of the J individuals that had previously visited that site?

Per R1's comment regarding the SIR notation in L203-206, I assume that this mean field notation is presented as a schematic, while actual simulations were run using the site-specific calculations described in the following lines. Could you provide an explicit link between the environmental decay rate \\alpha and the decay rate described on L212?

Change notation on L136 per reviewer's suggestion.

You may consider the reviewer's suggestions about citations and make changes if you feel they are helpful, but I will not require any changes at this stage.

Sincerely,

Matthew (Matt) Ferrari

Associate Editor

PLOS Computational Biology

Virginia Pitzer

Deputy Editor

PLOS Computational Biology

[LINK]

I thank the authors for their careful attention to the reviewer comments. Based on the comments of R2 I would suggest a few minor changes (detailed below) before recommending this manuscript for publication. With these changes made, I see no reason that this will need to be sent back out to review and I will be happy to direct the manuscript for publication. Specifically:

R2 raises a question about the notation on L216. I assume, from the text, that the sum described in that line is over individuals. However, R2 found that the description left some doubt. Could you please clarify this notation. Perhaps (a suggestion, not a requirement) adding an index j in 1...J to k_x,y to indicate the you are summing the decaying pathogen loads of the J individuals that had previously visited that site?

Per R1's comment regarding the SIR notation in L203-206, I assume that this mean field notation is presented as a schematic, while actual simulations were run using the site-specific calculations described in the following lines. Could you provide an explicit link between the environmental decay rate \\alpha and the decay rate described on L212?

Change notation on L136 per reviewer's suggestion.

You may consider the reviewer's suggestions about citations and make changes if you feel they are helpful, but I will not require any changes at this stage.

Reviewer's Responses to Questions

**Comments to the Authors:**

Reviewer #1: All my previous comments were excellently addressed. I have no further comments.

Reviewer #2: The authors have satisfactorily addressed the majority of my concerns and significantly improved the manuscript. However, some additional problems arise with the added material. Once they are fully addressed in the next revision, the paper could be a valuable contribution to the fields of both movement ecology and disease ecology.

Major problem:

1. Notations in the disease model section L203-219 are confusing and not always consistent. Is beta_p a constant as implied? If so, then it appears, based on L203-206, that the loss of susceptible to infection doesn’t depend on pathogen load P. Readers may thus wonder how pathogen load would even affect transmission dynamics. L214 suggests that beta_p is in fact a function of local pathogen load at time t, seemingly notated as k_x,y(t) – it would be more clearly denoted as k(\\emph{x},t). However, this dependency is not reflected in the main SIR description, where only the global pathogen dynamics is shown. As a consequence, the equations show no obvious coupling between individual movement and transmission.

L216 is especially confounding. 1) Does P(S->I) denote the probability of a single individual at location \\emph{x} and time t going from susceptible to infected, or some changes in global SI proportions as the notation might easily suggest? I assume the former. If so, describing this verbally is highly recommended. 2) What is k(\\emph{x},t) being summed over? Individuals? Since this variable measures the existing quantity of pathogen at a site, then it should already include remaining pathogen load from previous visitors, thus making the summation sign unnecessary. Having additional disease modelers read over this section and revise some of the notations may be very useful.

Minor problem:

L38-41: Two recent paper that are related to this study and may deserve discussion:

- the effects of disease dynamics on movement dynamics: Potts et al. Quantifying behavioral changes in territorial animals caused by sudden population declines. 2013

- the effects of movement dynamics on disease dynamics: Tao et al. Logistical constraints lead to an intermediate optimum in outbreak response vaccination. 2018

L113-116: This material related to population effect still seems beyond the scope of the analysis in this paper. I recommend removing these sentences.

L136: notation edit – P(a,b)

R2.8 – The model by Bateman et al. (2015) is more appropriately grouped with the other citations in the next sentence instead of with Okubo and Levin (2001).

**Have all data underlying the figures and results presented in the manuscript been provided?**

Reviewer #1: Yes

Reviewer #2: Yes

PLOS authors have the option to publish the peer review history of their article (what does this mean?). If published, this will include your full peer review and any attached files.

Reviewer #1: No

Reviewer #2: No
---

## [Editor Report · Decision Letter 2]

8 Apr 2020

Dear Dr White,

We are pleased to inform you that your manuscript 'A mechanistic, stigmergy model of territory formation in solitary animals: Territorial behavior can dampen disease prevalence but increase persistence' has been provisionally accepted for publication in PLOS Computational Biology.

Best regards,

Matthew (Matt) Ferrari

Associate Editor

PLOS Computational Biology

Virginia Pitzer

Deputy Editor

PLOS Computational Biology

I thank the authors for their careful consideration of the reviews and am happy to recommend the manuscript for publication.

---

## [Editor Report · Acceptance letter]

5 May 2020

PCOMPBIOL-D-19-01705R2 

A mechanistic, stigmergy model of territory formation in solitary animals: Territorial behavior can dampen disease prevalence but increase persistence

Dear Dr White,

I am pleased to inform you that your manuscript has been formally accepted for publication in PLOS Computational Biology. Your manuscript is now with our production department and you will be notified of the publication date in due course.

With kind regards,

Sarah Hammond
